# The Role of Moscow Patriarchs in the Promotion of the Imperial Culture of *Sobornost'*: Thematic Analysis of Religious Leaders' Speeches at the World Russian People's Council 1993–2022

**Alar Kilp** [1] and **Jerry G. Pankhurst** [2,*]

1    Johan Skytte Institute of Political Studies, University of Tartu, 51003 Tartu, Estonia
2    Department of Sociology, Wittenberg University, Springfield, OH 45504, USA
*    Correspondence: jpankhurst@wittenberg.edu

**Abstract:** In post-soviet Russia, *sobornost'* has been a historic ideal and cultural resource that diverse actors have used in order to construct anew the nation's dignity and status. This study analyses the promotion of the (imperial) culture of *sobornost'* by Patriarch Kirill and Patriarch Alexy based on 36 speeches they delivered from 1993 to 2022 at the World Russian People's Council, in a forum purposefully established to enhance the culture of *sobornost'*/*solidarity* in Russian society. The findings of a qualitative thematic analysis of the speeches identified common themes (such as 'true historical path'), thematic changes (such as the adoption of geopolitical discourse on family), thematic emphases uniquely present at particular 'times' (such as at the EU enlargement of 2004), themes related to the promotion of *sobornost'* at the level of the trans-national church, and its correlates—Russian state-civilization, globalization, and confrontation with the West. The findings demonstrate agreement in the messages of Patriarch Alexy and Patriarch Kirill as well as specific content and style that were articulated only by the latter. In the conclusion, we compare Kirill's culture of *sobornost'* with Roman Catholic synodality and with Russian 19th century applications of the same concept, and Kirill's entrepreneurial construction of national identity from the perspective of glocalization.

**Keywords:** World Russian People's Council; Kirill, Patriarch of Moscow & All Rus'; Alexy, Patriarch of Moscow & All Rus'; Russian imperialism; religious leadership; *sobornost'*; political culture; glocalization



## 1. Introduction

The attack by Russian forces on Ukraine that had its first battles in 2014 and entered its most vicious, violent, and intense phase with the all-out military invasion starting in February 2022, has stimulated a great movement of self-reflection and introspection for global Eastern Orthodoxy. The pro-war, pro-Putin leadership of Patriarch Kirill, primate of the Russian Orthodox Church, raises many questions about religious responsibility and Christian leadership. Conveniently, we have a 30-year record of the thinking of the two Russian Patriarchs who served during the post-Soviet period—Alexy II (Ridiger) and Kirill (Gundiaev)—and this record allows us to see the nature of the thinking of these major leaders and reflect upon the way this thinking may underlie Patriarch Kirill's support for the Russian invasion of the predominantly Orthodox Ukraine. Our analysis of the narrative record of speeches of the patriarchs indicates that a key concept upon which Kirill bases his pro-war public pronouncements is that of sobornost'. This term has a special role in 19th and early 20th Century Russian theology and social thinking, but our goal here is not to re-interrogate that discourse or associated theories of history. Instead, we focus on the construction of narrative by the two Russian patriarchs who led during the 1990s and into the present.

The concept *sobornost'* refers both to an institution—Council (*Собор*)—and to a particular pattern of collective cultural behavior and decision-making. In 1993, when the

World Russian People's Council (*Всемирный Русский Народный Собор*, WRPC n.d.) was established and had its first forum, the term had two roles: its title included the term Council (*Собор*) and *sobornost'* was the central concept of the theme of its first session, which was titled "Russian Councilial thought" (*соборная мысль*).

Scholars have defined *sobornost'* as one of main concepts characterizing Russian traditional culture (Biryukov et al. 1995, p. 150; Botz-Bornstein 2008, p. 844; Laruelle 2004, p. 30). Some even claim that *sobornost'* is the main feature of Russian culture and consciousness (Biryukov 1998), and the most 'original' concept of community Russians can think of (Botz-Bornstein 2008, p. 844).

So, what is *sobornost'*?

*Sobornost'* emphasizes the union of people based on common cause, common purpose, and inner motives (Stepin 2015, p. 184). The unity of *sobornost'* can be understood and applied from both bottom-up (Billington 2007) and top-down perspectives. *Sobornost'* has the following features:

1. **Holism**. Symbolically, it means that everybody is included, and nobody is left out. But unlike social democracy, for example, which also aims at inclusion of all within the framework of 'solidary society', *sobornost'* tends to disregard the distinction of any specific actors (with interests conflicting with the whole or with other members of the whole) within its single whole. In contrast, classical social democracy presumes that society consists of distinct social classes and distinct types of self-interested actors—such as government, trade unions, and employers' associations—who participate in the political decision-making process. *Sobornost'*, however, does not aim at the unity of autonomous parts, but at 'conciliar wholeness', where the whole is prioritized to the degree that all internal distinctions appear irrelevant (Biryukov et al. 1995, p. 151). In this fashion, *sobornost'* holistically unites the individual, society, and the state (Oushakine 2007, p. 190); it *assumes* the intrinsic unity between the rulers and the ruled (Biryukov et al. 1995, p. 153); it harmoniously balances authority and freedom (Turunen 2007, p. 325), and the individual person and the community.

2. The intrinsic unity of *sobornost'* is also **mystical** and **mythic**. Its vision of unity is suprapersonal and atemporal, similar to a religio-aesthetic consciousness rather than to a political unity (Botz-Bornstein 2008, p. 845). It presumes an intrinsic consensus to arise from the mythic unity of the social group (Biryukov 1998). Consequently, it leaves no room for deliberative procedures of decision-making where participant actors pursue dialectically conflicting agendas or the rules of decision-making and of accountability, attributing different roles for diverse actors. Bottom-up approaches to *sobornost'* may sustain and support liberal democratic procedures, but a top-down approach does not.

3. *Sobornost'* is quintessentially **Orthodox**, (some versions of it are, including the version of Patriarch Kirill) and particularly or uniquely **Russian**. In the Orthodox world, *sobornost'* may refer to 'conciliarity' and 'unanimity in freedom' (Ware 2011), 'religious sociality' and 'togetherness' (Agadjanian 2021, p. 12), 'communal solidarity', and unity in the communion of local Orthodox Churches (Naletova 2009, p. 380). In Russia, the originally religious principle has been transferred into the realm of political culture (and, as this study will demonstrate, religious leaders have been instrumental in this process). The principle functions simultaneously in the realms of both religion and politics. *Sobornost'* has become the unique Russian cultural version of community spirit and of "harmony in diversity", but the same principle also destines the Russian nation to sustain a great power state, an empire, which in turn realizes the values of both Russian spirituality and of *sobornost'* (Urban 1998, p. 980).

4. *Sobornost'* demands that **political decisions and actions be conducted "all in common"** (Biryukov 1998). The only legitimate political agency is community as a whole (Biryukov et al. 1995, p. 152). *Sobornost'* has "a preference for collective decision-making" (Pain 2016, p. 50). Its representative body (which in Russia includes the WRPC) 'represents' the community as a whole as an ideal model, as "a kind of unstructured unity that permits no internal divisions" (Biryukov et al. 1995, p. 152).

The term *sobornost'* was first used by Slavophile Aleksei Khomiakov (1804–60) as an antithesis both of Western individualism (Biryukov and Sergeyev 1993, p. 60) and the authoritarianism of the Roman Catholic Church (Engelstein 2001, p. 142). We find these themes present in the speeches of religious leaders in this study as well. Later Slavophiles—including Nikolai Danilevsky (1822–85)—developed the idea of Russia's "special path" (Pain 2016, p. 50), a theme more thoroughly and systematically elaborated by Patriarch Kirill than by Patriarch Alexy. Additionally, the *sobornost'* of both patriarchs overlaps with Nikolai Danilevsky's version of Russian particularism, which is expansionist and anti-European, and emphasizes Russia's uniqueness as a civilization (Tsygankov 2017, pp. 587, 591).

Unlike the versions of Alexy and Kirill, Khomiakov's *sobornost'* was exemplified mainly in the Russian village commune, the *obshchina,* the members of which constituted the *mir* (Copleston 2001, p. 75). In Khomiakov's view, the culture of *sobornost'* was functional in the traditional Russian peasant commune (*mir*) that combined its collectivism with a tradition of open, democratic participation and distributive justice (Sil and Chen 2004, p. 355). The latter is what we term the bottom-up or egalitarian-democratic version of *sobornost'*.

In 19th century Russia, the egalitarian-democratic and elitarian-authoritarian views of *sobornost'* overlapped in their emphases on "spiritual consensus" and in the use of the term *narodnost'* (Valliere 1978, p. 193). Even though the WRPC includes the term *narodnyi* in its name, in this study we did not find patriarchs to use the term *narodnost'* at all, although their messages are in a metaphorical 'symphony' with the official-imperial meaning that this term had in the 19th century (we elaborate this finding in the Discussion section).

The bottom-up and top-down views of *sobornost'* continue to co-exist at present. Patriarch Kirill, in this study, appeals to *sobornost'* and denies the autocephaly of Ukrainian Orthodoxy (in his very first speech at the first Council), which in 2019 significantly diminished the 'space of influence' of the Russian Orthodox Church (ROC) in Ukraine (Rousselet 2020, p. 46). Those in the Ukrainian Orthodox Church of the Moscow Patriarchy who disagree and want to have more autonomous and democratic decision-making and increasing independence from the Patriarch of Moscow, have also appealed to the same principle of *sobornost'* (Krawchuk 2022, p. 184).

Some have argued that after the Belovezh Accords ended the Soviet Union on 8 December 1991, the culture of *sobornost'* was in crisis in Russia (Biryukov et al. 1995, p. 156). It appeared that (democratizing) glasnost', openness and pluralism in the political realm of the late-Soviet Union and the pre-1993 Russian constitutional crisis violated the principles of *sobornost'* (Biryukov et al. 1995, p. 151). For a period of time, the top-down version of *sobornost'* seemed to lack both institutions and popular support.

In the mid-1990s, there were three main forces in Russian political society—democrats, patriots, and communists. They *all* asserted that Russia had lost its way, because vital connections with the past had been broken (Urban 1998, p. 969). Only democrats saw the ideal to emulate as outside of the nation's past (in the West). In contrast, both patriots and communists were committed to a degree of nativism (to traditions and historic culture) as a foundation for legitimacy and identity.

For example, Gennadii Ziuganov, who in 1993 became the chairman of the reconstituted Communist Party of the Russian Federation, also based his electoral campaign and popular appeal on "the thousand-year-old Russian tradition of *sobornost'*" (Laruelle 2011, p. 3; Vujacic 1996, p. 140). Again, when we trace Ziuganov's version of *sobornost'* in detail, we find elements that are missing in the discourse of the Patriarchs of Moscow. Ziuganov's construction of the Russian historic identity and values also emphasized *obshchinnost'* (Vujacic 1996, p. 149) (approximately "communality"), which is an element missing in the sample of speeches that we studied.

Consequently, in post-Soviet Russia, *sobornost'* was a historic cultural ideal, a common-sense cultural resource that diverse political and religious actors used in order to construct anew the nation's identity and dignity, values, and status.

This study focuses in particular on the speeches of religious leaders at the World Russian People's Council (WRPC n.d., *Всемирный Русский Народный Собор*), because this is the main institution in Russia where social and political issues are discussed with the participation of representatives of religion, in general, and in particular, the Patriarch of Moscow.

The WRPC is an international organization that emerged from an initiative of Kirill Gundiaev in 1993. It is the main think tank of the Russian Orthodox Church (ROC) (Chapnin 2020, p. 128). The aim of the WRPC is to be a forum where representatives of religion, politics, and civil society from both the Russian Federation and the Russian diaspora discuss spiritual, moral, social, and political issues and contemporary challenges. Kirill's motivating idea in establishing such a forum was to combine religious values with everyday life and to create the possibility for analyzing social and economic factors of life from a religious perspective (WRPC).

Between May 1993 and November 2022, 24 forums of the WRPC took place, the last of them in October 2022. Until the 13th WRPC (February 2009), Patriarch Alexy II acted as the honorary chairman of the Council and Kirill (at that time Metropolitan of Smolensk and Kaliningrad and the Chairman of the ROC's Department for External Church Relations) was his deputy. Since February 2009, Patriarch Kirill has headed the Council.

Each meeting of the WRPC has included a speech by the Patriarch of Moscow and also a speech by Kirill. We selected only official speeches and excluded from the transcripts of the Council meetings instances where Patriarch Alexy said the formal final words of the meeting, and where (Metropolitan) Kirill acted as a moderator of Council meetings or participated in Council discussions beyond his official speech. With this rationale of selection, we formed a collection of 36 speeches (12 speeches by Patriarch Alexy, 12 speeches by Metropolitan Kirill, and 12 speeches by Patriarch Kirill) and downloaded their content from the vrns.ru website in August 2021. Two speeches were exceptions: a speech by Metropolitan Kirill in the 7th Council, which is accessible only on the archival site of the Moscow Patriarchy (ROC Archive 2002) and the patriarchal speech at the 24th WRPC (October 2022), which was retrieved and added to the sample in November 2022.

For analytical purposes, we focus on 'three' speakers—Patriarch Alexy, Patriarch Kirill, and Metropolitan Kirill—in order to compare speeches delivered by two Patriarchs and by Kirill in two different offices.

Although messages of Metropolitan Kirill and Patriarch Alexy delivered at the same Council meeting are expected to overlap significantly, as well as messages by the same person (Kirill) in two formally different offices, such similarities should not be assumed without evidence. Mere count of words demonstrates a significant formal difference: the average speech of Patriarch Alexy was about three times shorter (642 words) than the average speech of Patriarch Kirill (2220 words). The average speech of Patriarch Kirill was about 2/3 in length of the average speech of Metropolitan Kirill (2928 words). In councils where both Alexy and Kirill delivered speeches, for every 20 words spoken by Alexy, Kirill delivered more than 90. And finally, despite Alexy having delivered one third of the speeches in the sample, his statements formed only about 10% of selected and coded quotes in the findings.

In analyzing the messages of their speeches, we used thematic analysis (TA) that identifies and interprets 'themes' within the text (Braun and Clarke 2006, 2019, 2021; Kuckartz 2014, pp. 69–87). By 'themes' we mean parts of texts which are identified and interpreted based on their meanings. As such, themes are not just 'abstract categories' or 'topics'—e.g., 'education' and 'culture' are such categories—which do not involve subjective meaning (Braun and Clarke 2021, p. 341). In contrast, 'tradition as a source of national unity' is a theme, which can be recognized in a speech due to 'repetition'" (Ryan and Bernard 2003, pp. 89–94) of the theme. And vice versa—themes are identified also by statements in the text that "go together because they have a common point of reference" (Lilja 2021, p. 1011).

Before analyzing the content of the speeches, we expected to find the following 'topics' in the speeches: religion, nation, identity, and values. Thus, in accordance with the TA approach, we as scholars were also part of the process of "generating (initial) themes" (Braun and Clarke 2019, p. 595). Our analysis of speeches—which in the stages that followed was 'exploratory' and 'inductive'—has been shaped by a research aim of our own—to identify messages related to the culture of *sobornost'* that are introduced and advanced in the Russian public sphere by Patriarch Kirill and two other 'speakers'.

We approached 'themes' not simply as ideas, but also as features of culture, which—in interaction with counter-themes—have a potential to control behavior and stimulate activities (Boutyline and Soter 2021; Opler 1945). Accordingly, to the extent that the message of the theme approves or denounces a certain type of behavior, these behaviors are representations of themes that express "the character, structure, and direction of the specific culture" (Opler 1945, p. 198). When themes are accepted and affirmed in a society, they will be translated into conduct or belief (Opler 1945, p. 199), becoming part of the 'social stock of knowledge', or a self-evident reality of everyday life that is taken for granted (Berger and Luckmann [1966] 1991, pp. 37, 56).

Following Udo Kuckartz (2014, pp. 70–84), Virginia Braun, and Victoria Clarke (Braun and Clarke 2006), we proceeded as follows: we read the speeches and highlighted paragraphs relevant to the four topics mentioned above; we generated initial codes; collated codes into themes; formulated summaries for themes; ultimately organized findings into three dimensions: 'years', 'speakers' and 'themes'; and summarized the results.

## 2. Results

The findings from the thematic analysis of the speeches are organized into five sections. Firstly, we present themes that were present in speeches without major changes over the years and without significant differences among the speakers. The second part presents themes and emphases of themes that have changed over the years. The third section highlights themes and emphases that 'popped up' during particular Councils and which can be interpreted as themes relevant for the particular time and context. The fourth part reviews the findings from the perspective of the promotion of the (imperial) culture of *sobornost'*. The final section is devoted to the discourse of religious leaders on Russian civilization, globalization, and confrontation with the West.

In the presentation of the findings, speakers are referred to in an abbreviated form (PA = Patriarch Alexy; PK = Patriarch Kirill; MK = Metropolitan Kirill), the in-text references include the year and the number of the WRPC meeting in brackets. Thus, the speech of Patriarch Alexy in the first meeting of the WRPC is referred as (PA 1993, 1).

WRPC meetings occurred in time as follows: 1993 (1), 1995 (2), 1995 (3), 1997 (4), 1999 (5), 2001 (6), 2002 (7), 2004 (8), 2005 (9), 2006 (10), 2007 (11), 2008 (12), 2009 (13), 2010 (14), 2011 (15), 2012 (16), 2013 (17), 2014 (18), 2015 (19), 2016 (20), 2017 (21), 2018 (22), 2019 (23), and 2022 (24). A chronological list of council titles and links to transcript sources of selected councils is included at the end of the paper (Appendix A). The links in Appendix A are to the webpages where all transcripts of the related Council are available.

### 2.1. Common Themes

Themes that are repeated and have not changed substantially over the years (although they are discussed more extensively during some councils and by some speakers) are 'unity', 'solidary society', 'true historical path', 'Orthodox values', and 'nation'.

#### 2.1.1. Unity

'Unity' is a theme that is particularly relevant during the first three Councils (1993 and two councils of 1995). 'Unification' around values is described to be the basis of the very existence of society (PK 2011, 15). Unity is not the result of pragmatism, it is "organic in relation to our national identity" (MK 1995, 2). The ROC contributes to the historically

established 'unity' of peoples belonging to the Russian civilization (Ukrainians, Belarusians, Moldovans, and "many others") (PK 2011, 15).

### 2.1.2. Solidary Society/Society of Sobornost'

The WRPC, as an institution, was designed to enhance *sobornost'* (PA 1995, 3) according to the social ideal of "solidary society (*солидарное общество*) . . . a society that is based on mutual assistance, on cooperation" (PK 2018, 22). When Alexy says that *sobornost'* represents an ideal, where everyone in Russia works "for the common good, perceiving any work as a service to the Lord and the Fatherland" (PA 1995, 3), the reference is to the same ideal Kirill has in mind, when he uses the term 'solidarity'. Russia is a 'solidary society', "a society of social symphony, where different groups, peoples and communities are not competitors fighting with each other, but co-workers" (PK 2013, 17). Solidarity is one of the basic or fundamental values of (Russian) national identity (PK 2011, 15). 'Solidary society' represents values that are shared by peoples who belong to Russian civilization (PK 2015, 19).

### 2.1.3. True Historical Path

The 'true historical path' involves historical experience that is not to be forgotten and historical fate that is to be followed in the present and future (PK 2017, 21). This theme is more thoroughly discussed by Kirill than by Alexy, yet there is no disagreement between the two. The theme is presented with a premise that unity of the people can be maintained only through a common understanding of its history, and a common understanding of history is the basis of the values and cultural code of the nation (PK 2014, 18).

Alexy claimed in the mid-1990s that Russia's 'historical path' had to be corrected by "the return to the paternal faith" and "to original spiritual traditions" (PA 1995, 3). Kirill, however, was more explicit. He identified positive examples—or 'proper historical choices'—in Russian history, which should be taken as an example to follow, as well as negative examples to avoid.

Proper and positive historical examples are as follows: the *civilizational* choice of St Vladimir in 988 (PK 2015, 19) to opt of the Christian religion and the *historical* choice of Russian rulers during the XV century not to become a nation-state typical of Europe and to instead choose universalism (Kirill is careful with the word 'empire', but here it is fully obvious that the reference is to empire) instead (MK 1993, 1). Other examples include '1612' (symbolized in speeches as a period of "Troubles", "*Смута*") just before the first Romanov tsar was enthroned and saved Russia from Polish control; '1812' (also symbolized in speeches by the term Borodino) when Napoleon's army was defeated near Moscow; the Second World War (symbolized in some speeches by numbers '1941' and '1942' or with the word Stalingrad). All of these examples involved confrontations with the West for defense of either *religious* (1612), *cultural* (1812), or *physical* (1942) independent existence. Kirill refers to 1612, 1812, and the Second World War in several of his speeches (MK 1993, 1; MK 2005, 9; PK 2012, 16; PK 2014, 18; PK 2016, 20) as instances where Russians historically "defended our right to life, liberty and independence" (PK 2016, 20), while Patriarch Alexy does not refer to Kulikovo or to years 1612 and 1812 (or Borodino) at all. Additionally, Kirill's degree of utmost commitment to the secular state is a style not expressed by Alexy:

> "Love for the motherland, a sense of brotherhood and a sense of duty, a willingness to lay down "one's soul for one's friends" are equally characteristic of the heroes of Kulikovo Field, Borodino and Stalingrad. These same qualities of national character distinguish most Russian people today." (PK 2014, 18)

Self-sacrifice for defense of the Motherland is praised by both, but only Kirill goes as far as:

> " . . . it is better to perish (*лучше погибнуть*) defending the Motherland (*Rodina*) by force of arms than to allow it to fall apart or be enslaved. It means that in our society there are values that morally exceed the price of our lives." (MK 2001, 6)

Historical examples are considered negative when they either contributed to divisions and disintegration of Russia or involved a rejection of spiritual ideals and traditions in favor of Western ones. Similar to the October Revolution, the Pugachev rebellion was promoted by an army of "assimilated internationalists, whose members are torn from their roots, do not know either clan or tribe . . . [are] without holiness in the soul" (MK 1993, 1).

Peter the Great (1682–1725, tsar 1682–1721, from 1721 first *Emperor of All Russia*) is a negative example, due to the enslavement of the consciousness with alien ideas and alien models that occurred during his rule (MK 1995, 2).

The Soviet regime devastated Russian people spiritually, burdened "people's hearts with untruths and sins" (PA 1995, 3), and replaced the true Orthodox principle of universalism with internationalism that put the Russian nation in a lower position compared to others (MK 1993, 1). In particular, the event of the October Revolution of 1917 (MK 1997, 4) represents a "revolutionary catastrophe" (PK 2017, 21), "a tragedy" resulting from political radicalism (PK 2017, 21), a "temptation of unbelief and state nihilism" (MK 1993, 1).

The most critical interpretation of the 1990s is found in speeches delivered in the 21st century (PK 2011, 15), although the perception of crisis was also acute *during* the 1990s. Metropolitan Kirill said in 1995:

> "Whatever sphere of life we take, there is a crisis. Politics is a crisis, economics is a crisis, ecology is a crisis, culture is a crisis, the army is a crisis. . . . This is a crisis of personality, a crisis of our self-consciousness". (MK 1995, 2)

The 1990s have also been characterized as a period which passed "under the sign of destruction and chaos" (PK 2014, 18), a period where economic principles ruled over spiritual values (PK 2009, 13), Russian history was being revised (PK 2014, 18), the meaning of the word "Russian" was being distorted (PK 2014, 18), and historic Russia was disintegrated:

> "Our neighbors in the CIS have clung to nationalism as a level of self-awareness and self-realization. And in the face of these local nationalisms, the Russians were defenseless. The principle that binds the nation has been lost. "Russians" became an empty phrase, denoting nothing by name." (MK 1993, 1)

Problematic periods of Russian history also have overlapping characteristics—the Pugachev Rebellion, the October Revolution, and the 1990s involved 'assimilated consciousness', which undermined Russian culture and moral traditions (MK 1993, 1).

### 2.1.4. Orthodox Values

'Orthodox values', or values supported by the Orthodox Church, are an indispensable basis of all dimensions of Russian social, cultural, and political life, as well as the foundation upon which Russia as a state and civilization acts or should act in the modern and global world.

The speeches of Patriarch Alexy during the first three councils emphasized the natural connection between the service of God and Fatherland: "Christian patriotism is sustained by Christian love for Fatherland (*Otechestva*)" (PA 1993, 1); sincere faith needs to be in the heart of every Russian person (PA 1995, 2); everyone in Russia should perceive "any work as a service to the Lord and the Fatherland (*otchizne*)" (PA 1995, 3).

God and Fatherland also remain themes in other speeches, but their connectedness is later taken for granted. The same applies to the frame of the 'Holy Rus", which was outlined and explained in the first Councils (PA 1993, 1; MK 1993, 1; MK 1995, 2) and mentioned as a phrase only twice in later speeches (PA 2001, 6; PK 2013, 17). In the mid-1990s, both Alexy and Kirill primarily emphasized the religious and spiritual dimension of the concept, but Kirill approached it from the perspective of the (great and strong) nation:

> " . . . a new Rus' will rise from the ashes and from the abyss of sin—Rus', which gave the world many ascetics of piety, Rus', which creates temples in cities, villages and hearts, Rus', shining to the whole world with truth and love, Holy Rus'.". (PA 1993, 1)

"The churching of our people, the return of the prodigal son to his father's house—only this can resurrect Russia and give it the strength to become Holy Rus' again, capable of resisting the forces of hell." (MK 1993, 1)

Orthodox values listed in several speeches are predominantly secular in character (i.e., not those which are conventionally related to institutional religious belief, identity, and practice). In 2011, Patriarch Kirill listed the fundamental values of national identity as follows: "justice, peace, freedom, unity, morality, dignity, honesty, patriotism, solidarity, family, culture, national traditions, human welfare, diligence, self-restraint, sacrifice" (PK 2011, 15). In 2004, he said that 'values' supported by the ROC are "a strong and independent state, a free and responsible individual, an effective and fair economy, protection of the weak, a strong family, respect for authorities, honest work" (MK 2004, 8).

In particular, when the speeches discussed the Orthodox approach to 'personal gain', work ethic, and 'economic justice', the message was social and cultural (not explicitly religious) in character. Thus, "selfishness and indifference to someone else's misfortune" needs to be abandoned (PK 2017, 21), 'pursuit of pleasures' and 'desire for profit' are vices (PK 2009, 13), and the economic principle of benefit should not be elevated over the spiritual values and Christian moral norms (PA 2002, 7). The only proper motivations for the Orthodox conception of labor are "to work for oneself and for those in need" (MK 2002, 7).

The growing gap between rich and poor countries in the world or between the rich and poor in Russia are not in agreement with the "sense of justice" that is "an integral part of Russian identity" (PK 2018, 22). According to Russian Orthodox traditions, "wealth was justified only if it was used for the benefit of all" (MK 2002, 7), wherefore "the [r]ich should share with the poor the surplus of their fortune" (MK 2002, 7). In contrast, the Soviet system was unjust, because it alienated "man from the results of his own labor" (MK 2002, 7). In 21st century society, poverty can also be sinful, because "poverty can become a strong temptation for a person, embitter him, plunge him into the depths of despair and even push him to a criminal path" (PA 2007, 11).

With 'Orthodox values' one meets and tackles 'problems of modernity', geopolitical relations, integration of peoples who belong to historical Rus', globalization, or relations between East and West. Such problems of modernity are: "dirt and lies that pour from television screens, from the internet, from the pages of the tabloid press" (PA 2008, 12); individualism (MK 1993, 1; MK 2006, 10) and "the inability of society to unite" resulting from extreme urbanization (PA 2006, 10); and the 'cult of enrichment' as the main goal of life "preached from departments, political tribunes, television screens, newspaper and magazine pages" (PA 2007, 11).

Until the Patriarchy of Kirill (from 2009), some 'problems of modernity' seemed to have been more intensely present in Russia than in West. For example, economic and social injustices in Russia have been among the reasons, why "[m]any capable and hardworking people go abroad in search of more favorable working conditions" (PA 2007, 11).

Kirill (in his speech at the same 2007 council) shares the view that the challenges of modernization are the same everywhere. Modernization as a process "has external attributes that are the same for all peoples: these are good roads, airports, means of transport, the welfare of citizens, a high level of science and education" (MK 2007, 11); moral vices are being promoted by "cinema, television, video, and especially computer programs" and offer "as the most radical and the fastest and easiest and most beautiful solution to any problem" (MK 1997, 4). More recently he listed 'belief in technology' as source of 'problems of modernity':

"Belief in technology . . . is also a kind of quasi-religion . . . is a person's belief that with the help of science and technology it is possible to achieve perfection and immortality, complete power over one's body, over nature, over life. But that's not possible... All this leads away from the main Christian path. Ultimately, in the direction of dehumanization, hypertrophied individualization, and hence the destruction of society and the end of history.". (PK 2017, 21)

Part of these modern problems, however, emerge not from inevitable processes of modernity, but from the ideology of radical secularism, which promotes abortions, undermines the institution of family, erodes basic moral values, and aggressively attacks traditional religious cultures with the aim of large-scale and purposeful de-Christianization (PK 2016, 20).

2.1.5. Nation

'Nation' is a theme developed in the speeches with five different meanings, some of which are in contradiction with each other, but superficial logical contradictions cease, when we carefully trace the application of different meanings for different cases.

First, Russia and Orthodoxy 'are national'. Russian Orthodoxy adds a universal dimension to people's life, but it is also 'deeply national' (PA 2001, 6).

Second, Russia and Orthodoxy are universal and imperial, not national. In the first Council, Metropolitan Kirill advocated for the revival of Russian imperial universalism and denounced the rise of political nationalisms in former states of the USSR:

> " . . . in the conditions of growing nationalism of small and medium-sized peoples who made up Russia (the former USSR), the attempt to establish Russian universalism will be perceived as a kind of imperial ambition of the "big brother" or "great neighbor". . . . This revival must be sensitive to the national feelings of others, it cannot be aggressive." (MK 1993, 1)

Third, Russia and Orthodoxy are multinational, wherefore the true Orthodox person is not a nationalist (MK 1995, 2). Over the centuries, different cultures and religions have retained their religious identity and coexisted in Russia by virtue of sharing the same values of social life (MK 2004, 8). Christianity is a universal religion, each nation has a God-given opportunity to express its national face, and a peculiarity of Russian civilization is that there are no people that are superior over others (PK 2015, 19).

Fourth, there is a kind of nationalism not characteristic of Russia and Orthodoxy (the exemplary case of such a negative relationship is Ukrainian nationalism). A positive example of Orthodox patriotism existed during the Second World War, when the ROC acted in the spirit of 'sacrificial patriotism' in contrast to 'narrow nationalism', which acts based on racial or class hatred (PA 2005, 9).

While the Russian people have (arguably) "never been characterized by xenophobia or intolerance on a national basis" (MK 2005, 9), it is Ukrainian nationalism that has elevated nationalism "to the highest principle," and has legitimized the struggle that tears apart humanity and destroys "the brotherhood of the peoples who made up and make up Russia" (MK 1993, 1). In the case of Ukraine, religion has also been instrumentalized by national forces in an unjustified way. In 1993, Kirill already observed that: "There is a danger of 'Ukrainian autocephaly' emerging, a kind of religion which does not extinguish the "fighting spirit" of the nation" (MK 1993, 1). Kirill assumes that good religion does not stir up nationalist spirit.

Fifth, the fault of Ukrainian nationalism lies in the violation of the unity of Russia (PK 2014, 18), wherefore observations regarding nationalism in Ukraine do not apply to nationalism in other Orthodox countries. Russians, Ukrainians, and Belarusians are special cases, because they form the Russian people, because they all are heirs of Kievan Rus' (PK 2019, 23). The recent ('internecine') conflict in Ukraine has been a threat to the unity of Russia, because Ukraine lies in heart of historical Rus' (PK 2014, 18).

*2.2. Thematic Changes*

2.2.1. From Crisis of Modernity to the Crisis of Russia's Position in the World

During the 1990s, Patriarch Alexy's speeches did not include the word 'West' (he first used this term in 2001). For him, the main source of changes and difficulties was 'modernity' (PA 2004, 8). In the context of modernity, the whole world experiences crises of worldview and morality (PA 2001, 6).

It was in 1999, that Alexy for the first time addressed the question about the 'place of Russia in the world'. He argued that Russia needs to remain itself but also define anew its place in the world (PA 1999, 5).

A shift from 'problems of modernization in the world' towards 'problematized status of Russia in the world' can be found in the 2006 speech of Patriarch Alexy, where he focuses on the conceptualization of human rights in modern international law and asks to what extent it allows the Orthodox people to live in accordance with their religion:

> "Are we not on the verge of a new pagan renaissance, in which a firm standing on the stone of the faith of our ancestors will be universally recognized as undesirable?" (PA 2006, 10)

### 2.2.2. Tensions between Traditional Culture and Liberalism of Modernity/West

Kirill's speeches include a similar shift with the difference being that he also problematizes the West(ern human rights). In the same (2006) Council, Metropolitan Kirill also addressed the notion of human rights and argued that this concept is Western in origin, it has its successes and shortcomings (demographic decline, antisocial, and immoral behavior that accompanies the increased level of individualism in the West). Other civilizations and "every nation" have their own experience of social life, wherefore they have the right to determine the standards of human happiness for their country and culture on their own (MK 2006, 10).

The values of democracy, humanism, and human rights remain important in Russia, but these will be based on Russian tradition and culture, not on abstract global standards (PK 2017, 21).

### 2.2.3. Advent of Vladimir Putin

Another thematic change occurred after Vladimir Putin became president of the Russian Federation on December 31, 1999. (As there was no council in 2000, the first council during Putin's presidency occurred in 2001.) Patriarch Alexy's speeches started to focus on injustices in the modern world order (the uneven distribution of power and wealth among countries and continents) (PA 2001, 6). He called upon Russia and the Orthodox civilization to become one of the centers of decision-making in the world (PA 2001, 6). Instead of civilizational confrontation, however, he saw that the civilizational role of Russia was to be a 'bridge' that connects "East and West, North and South" (PA 2001, 6).

Again, Metropolitan Kirill made similar arguments, but more elaborate in content. For Kirill, Russia represents an original civilization which has equal right to determine the fate of mankind as Western countries (MK 2001, 6). At the time, when secular liberal humanism has expanded and raised the earthly interests of sinners over religious and moral values, the world-order must be 'multipolar' at the level of cultural and civilizational values (MK 2001, 6), and each country should be able to influence the adoption of important global decisions and "to live according the values of its people" (MK 2004, 8).

Additionally, absolutization of the sovereignty of the individual and his rights leads to the defense of abortion, homosexuality, euthanasia, and blasphemy, and can destroy modern civilization (MK 2006, 10).

In our multipolar and multi-civilizational world, there are multiple kinds of globalizations, wherefore common moral consensus as the basis of globalization can be found through inter-civilizational dialogue (PK 2018, 22).

### 2.2.4. True Clash of Civilizations

From the beginning of the 21st century, there emerged also a discourse around the 'true clash of civilizations', which positioned the battle lines *not only* between states and civilizations but also inside civilizations. According to Kirill, the true clash is manifested not between civilizations, but in "the conflict of traditional values, including religious ones, with secular-humanistic ones" (MK 2001, 6). Particularly in the speech of 2016, he emphasized that "there is another America and another Europe", which desires "to

preserve their Christian roots and cultural traditions" and "demand for a return to moral values, including Christian ones", wherefore the true clash "takes place not only along the borders separating states and regions, but also within countries and peoples" (PK 2016, 20).

While Western modernity included at least some positive examples to follow in the 1990s and was a partner of dialogue in the beginning of the 21st century, in 2016 Kirill argues that the path forward is no longer represented by the West: " ... this approach of 'catch-up development' can hardly be called the national interest; moreover, the very principle of 'catching up' *a priori* implies backwardness. If we catch up, we always lag behind" (PK 2016, 20).

2.2.5. Introduction of (Geo)political Discourse on Family

Finally, a particular change has occurred in the role of 'family' in the speeches of patriarchs. The word 'family' occurred in Patriarch Alexy's twelve speeches only four times, while in twelve speeches of Patriarch Kirill (starting from 2009) it occurred over sixty times. Obviously, the focus on family has become one of core themes regularly present in the messages of Patriarch Kirill. Family (involving both motherhood and fatherhood) is the basis of society (PK 2017, 21). Family is important for the demographic preservation of the Russian people (PK 2019, 23). Family is a source of mutual understanding, assistance, and happiness (MK 2008, 12). Society is a "family of families" and is threatened by the same forces that threaten the family: "the extremes of juvenile justice, same-sex marriage, the assertion of transhumanism, any attempts to give a distorted definition of the concept of 'man'" (PK 2017, 21).

Kirill's doctrine on conservative family values should be assessed within the collective project on the transnational family movement linking Evangelicals—in particular the Billy Graham Association—, Catholics, and Orthodox people (Shishkov and French 2017). In 2016, Patriarch Kirill and Pope Francis met in Havana, delivered a joint declaration, which claimed that "[T]he family is the natural centre of human life and society. We are concerned about the crisis in the family in many countries. Orthodox and Catholics share the same conception of the family ... " (Catholic News Agency 2016). The strategic alliance of Catholics, Orthodox people, and Evangelicals opposes the promotion of the rights of same-sex couples, aims at protection of traditional values, and promotes the pro-life agenda regarding abortion (Stroop 2016).

*2.3. Emphases in Particular Times and Contexts*

2.3.1. Churching of Culture

Speeches in the first three Councils (1993, 1995) highlighted the existence of a spiritual vacuum and the need for churching (*воцерковление*). In the 1990s, the problematic foreign actors (or 'foreign others') involved cults, sects, and missionary movements defined and treated as "destructive sects and pseudo-mission organizations" (PA 2001, 6) or "pseudo-religious garbage" (MK 1995, 2), not Western states conceived of as "powerful of this world" (*сильные мира сего*), as has become the case since 2014 (discussed in detail below).

Churching aims at 'churching of culture', which is a prerequisite for the resurrection of Holy Rus' in its full strength (MK 1995, 2). Churching takes place not only in the sense of the "religious revival of the people" (MK 1993, 1), but also in transferring the moral ideals of the Orthodox Church to personal, family, and public life, to intelligentsia, scientists, and all professional groups (MK 1995, 2).

Thus, churching is expected to fill the vacuum of spiritual and moral values that was a consequence of "long-term apostasy" (PA 2001, 6). However, in later years churching as a theme was less present and the extent of explicitly religious terms used in speeches declined. The word 'sin' (*грех*) was used by Patriarch Alexy three times and by Metropolitan Kirill 21 times, yet from 2009 until 2019 Patriarch Kirill never used that word and only twice mentioned the word 'sinful' (*грешно*) when talking about teaching a child morals, acts such as breaking a tree branch or polluting nature (PK 2009, 13), that are not exactly sinful in a strictly biblical sense. Therefore, it may be speculated that to the degree that 'churching' of

culture was transformed from a need to a social reality (and therefore ceased to be a theme that needed to be raised in Councils), there has occurred a parallel change manifested in the abandoning of words with explicitly biblical context (such as 'sin' and 'sinful'). In the most recent speech, delivered in October 2022, Patriarch Kirill mentioned sin three times when he blamed (the Western type of) secularization of society, which tends either to recognize sin as a virtue or lacks the concept of sin in its secular consciousness (PK 2022, 24).

### 2.3.2. Obedience to Political Authority

The Council of 1999 (6–7 December, the beginning of the month when Vladimir Putin became president of the Russian Federation upon the sudden resignation of Boris Yeltsin) is unlike other councils because of the explicit focus on 'power' (*Власть*) and authority. At this Council, the Alexy and Kirill's approaches to power are markedly different. Patriarch Alexy condemned pre-election behavior of some political candidates—the actual election was scheduled for the following spring—and highlighted the following problems of the political sphere: the political administration is inattentive to the common man; corruption; and concentration of power in the hands of very few people (PA 1999, 5).

Metropolitan Kirill spoke of the biblical attitude to power, which considers "power as a necessity, in order to ensure and organize the life of society, the people and later the state" and where secular power without spiritual authority cannot be legitimate (MK 1999, 5). At the time of the Council, Kirill asked:

> "What is happening in the mass consciousness? In the mass consciousness there is a certain disregard for the institution of power, a certain contempt for people who are running for power, who may be in power tomorrow." (MK 1999, 5)

The man to be in power "tomorrow" (three weeks after the Council) was Vladimir Putin.

### 2.3.3. The 2004 Enlargement of the European Union as a Critical Juncture

The enlargement of the European Union into east-central Europe in 2004 was another critical juncture considered in detail in the speeches. Patriarch Alexy highlighted the way that relations with the Orthodox countries in this region had changed since the period of Communist regimes. In the Soviet period, the relationship was molded by a policy of domination, but now it should be "fraternal service of the Orthodox peoples to each other" within the framework of unity of the Orthodox peoples as an independent civilization (PA 2004, 8).

Metropolitan Kirill added that states of the Orthodox tradition include Bulgaria, Belarus, Greece, Cyprus, Moldova, the Republic of Macedonia, Russia, Romania, Serbia and Montenegro, and Ukraine, and noted Orthodox minorities among Albanian, Polish, Czech, Slovak, Finnish, and other peoples, wherefore:

> " . . . modern Russia seeks to build the closest relations with the European Union and with the Western world in general. This policy of openness and cooperation with the West speaks of the coincidence of the main vector of foreign policy of all countries of the Orthodox tradition." (MK 2004, 8)

The attitude regarding the European Union was unusually positive during the Council of 2004.

### 2.3.4. The 2012 Presidential Election Amidst Another 'Crisis of Values'

The Council of 2012 took place in October, ten months after very contentious Russian legislative elections late in 2011 that initiated a wave of political protests, that were unprecedented since the 1990s, and continued into 2013. In February 2012, the punk group Pussy Riot carried out a scandalous disruptive action on the altar of Moscow's Christ the Savior Cathedral that made global headlines and further destabilized the symbolic balance of authority in the ROC and society. And in April, Vladimir Putin was re-elected for the third term as president, gaining the office back from Dmitry Medvedev who had served one term in alliance with Putin in the Prime Minister's chair.

Against this background, Patriarch Kirill spoke once again about the 'vacuum of values' (defined as a situation where people have nothing to give their lives for or to self-restrain themselves) which should be filled with fundamental values of national identity such as "justice, peace, freedom, unity, morality, dignity, honesty, patriotism, solidarity, family, culture, national traditions, human welfare, diligence, self-restraint, sacrifice" (PK 2011, 15).

In the Council of 2013, he emphasized that " . . . the denial of values has become one of the most dangerous manifestations of the forces of spiritual destruction. In this situation, the protection of values is the protection of our spiritual sovereignty" (PK 2013, 17).

2.3.5. November 2014—'Powerful of This World' Have 'Turned against' Russia

For Kirill, the change of the Ukrainian government in 2014 resulted in an " . . . internecine (*междоусобный*) conflict" taking place "in the heart of historical Rus'—in Ukraine" (PK 2014, 18). By denying subjectivity and agency to the Ukrainian political nation, Kirill contributed to the transformation of the image of Ukrainians from 'little brothers' to dehumanized others that took place in the Russian public media from 1 November 2012 to 31 October 2014 (Khaldarova 2021). During this time, Russia illegally annexed Crimea and promoted revolt in the eastern Donbass region that installed rebel governments in two territories.

In speeches from 2016, and up to 2019, Kirill uses the phrase 'powerful of this world' five times, which can be interpreted in three ways: 1. A certain re-positioning of 'self' has taken place, as the phrase would be meaningless if the perception of Russia and the ROC in the world would be that of a participant that is of equal status to other 'main players'; 2. The use of this phrase may refer to a degree of a sense of powerlessness, whereby instead of being able to act and achieve what is proper and due (historically), there is a perception of being acted upon by forces that are powerful in this world; 3. The discourse of a 'victimized' self has been also articulated at times, when the Russian Federation has de facto pursued politics of aggression against its neighbor (Ukraine). Therefore, it may be instrumentally used for legitimization of aggression and not be a response to the domination from the disempowered victim.

According to Kirill, the 'powerful of this world' have turned against the ROC and Russia in an attempt to tear off the Greek Orthodox world from the Russian Church, to destroy the unity of the Orthodox Church, and to try to impose world order by force and money that is not suitable to those "who do not want to lose their identity and their sovereignty" (PK 2019, 23).

The same attitude already existed in November 2014, when Patriarch Kirill said that western powers dominate "the global information space, they impose on the world their understanding of the economy and the state system, seek to suppress the determination to defend values and ideals that are different from their values and ideals" (PK 2014, 18).

Kirill's assessment of both present and future is gloomy. Ideological confrontations in the world are growing (PK 2018, 22), the value gap between Russia and the countries of Western civilizations has grown to a degree that did not exist even during the Cold War (PK 2016, 20), and the global struggle is over the basics, over the definition of what a 'person' is (PK 2017, 21).

His main concern is the unity of the civilizational space, which "largely coincides with the canonical territory of our Church" (PK 2018, 22), and the struggle for the preservation of the unity of Universal Orthodoxy and the spiritual independence of the Russian Orthodox Church in the face of 'the powerful of this world', who want to destroy the unity of the Orthodox Church (PK 2019, 23).

*2.4. Promotion of the (Imperial) Culture of Sobornost'*

2.4.1. Two 'Crises' of Russian Imperialism in 20th Century

In the very first sentences delivered in the first Council, Metropolitan Kirill identified two periods of crises—1917 and 1990s:

"Any reflections on the former greatness of the Russian state inevitably rest on the question: how could it happen that the centuries-old mighty statehood was crushed. This crushing began in 1917 and seems to continue to this day." (MK 1993, 1)

In later speeches, Kirill labels 1917 and the 1990s as 'tragic events', 'catastrophes', periods of 'chaos', 'destruction', and 'turmoil' (MK 2004, 8; PK 2012, 16; PK 2015, 19; PK 2017, 21; PK 2018, 22). Part of the blame is attributed to the national elites, who were, during both 1917 and the 1990s, fascinated "with ideas that have no roots in Russian reality" (PK 2017, 21).

When Russia collapsed (the collapse of the former USSR is constructed as the collapse of Russia), the Russian people found themselves in a humiliated position, their self-identification was largely lost (MK 1993, 1), and self-awareness was in crisis (MK 1995, 2). By this reasoning, the standard of normality is the historic imperial Russia.

According to Kirill, the 'Orthodox-national principle' that Russian society has to follow is "the idea of the Russian oecumene" and to once again reach the 'universal ideal' of 'Russian universalism' (MK 1993, 1). In Kirill's vocabulary, 'universalism' is a defining trait distinguishing Russia from the typical European nation-state. Thus, universalism (and by implication empire) is the only way for Russia. This is Russia's calling, this is the plan that is in accordance with the nature of the Russian nation (MK 1993, 1).

The first message of Patriarch Alexy similarly emphasized the need for unity:

"The Russian Orthodox Church has been and will remain a force that contributes to the formation of the national spirit of Russia, creates its spiritual culture, cares about its unity and integrity as a national-state organism." (PA 1993, 1)

However, Alexy had an essentially religious view of the crisis Russia was in. According to him, the new Holy Rus' will need to rise from the "abyss of sin" (PA 1993, 1). Here and in many instances later, despite significant common takes on many issues, Alexy tends to retain a religious, rather than a secular focus. For Kirill, secular concerns are often above religious ones, despite the claim that "the norms of religious tradition are more authoritative for the believer's consciousness than earthly laws" (MK 2006, 10). In the latter case, his lists of "the norms of religious tradition" tend to be focused on general norms of (secular) social ethics (such as honesty).

### 2.4.2. Promotion of Sobornost'

The culture of *sobornost'* is promoted by both Alexy and Kirill. In 1995, Alexy affirmed that "today Russian society lacks genuine conciliarity (*подлинной соборности*)" (PA 1995, 3). The WRPC as an institution was established to initiate a 'conciliar mind' in the national life of the Russian people (PA 1995, 3). In this institution, deliberation is conducted "with a conciliar mind" (*соборным разумом*) (MK 2006, 10) and "the conciliar movement" (*соборное движение*) takes place (MK 2004, 8).

As the principle of *sobornost'* (in its top-down version) presumes that the single collective deliberates and decides consensually 'all in one', the messages of religious leaders should therefore not (sic!) be discerned from any other voice and speaker. Accordingly, it is in accordance with the culture of *sobornost'* and the purposes of the WRPC, when Metropolitan Kirill said: "not a single speech at this Council should be identified with the official position of the Russian Orthodox Church" (MK 1995, 2).

According to Kirill, the WRPC as an institution is "a space for a national discussion about the fate of the Russian people and Russian statehood" (MK 1995, 2), "an instrument of consolidation of the nation" (PK 2022, 24), a public forum that unites "people of different generations, vocations, worldviews, political views . . . around the idea of serving the people of Russia" (PK 2011, 15), and contributes "to the unity of all Russian people and expressing their true aspirations and beliefs" (PK 2014, 18). The WRPC promotes the dialogue that "unites all parts of our society with one solidary aspiration—love for our Motherland" (PK 2017, 21).

Is the *sobornost'* of Kirill identical with the *sobornost'* of Alexy? Yes and no.

Kirill defines *sobornost'* as a principle "which harmonizes the principle of social unity and personal freedom through love for God and neighbor" (MK 2004, 8). With this definition, both bottom-up and top-down versions of *sobornost'* agree. It may be speculated that, in order not to be identified with the 19th century bottom-up Slavophile versions of *sobornost'*, he refers to *sobornost'* without reference to the previous formal meetings of the institution in his 2009 address to the Council, where he speaks about 'conciliar thinking' (PK 2009, 13). In later speeches, he uses the ideal of solidarity and of 'solidary society' as referent concepts instead.

To our surprise, we found the transcripts of Patriarch Alexy's speeches do not include terms such as 'solidary society' and 'solidarity' at all.

Thus, in the office of Patriarch (since 2009), Kirill affirms that the WRPC as an institution promotes the ideal of 'solidary society' (PK 2012, 16; PK 2013, 17; PK 2015, 19). It is now 'solidarity' that has defined "the entire historical path of Russia" (PK 2013, 17), that guarantees the unity of the nation (PK 2012, 16). Now 'solidary society' is "a Russian social ideal" (PK 2015, 19), a society of unity and brotherhood (PK 2017, 21), of "mutual assistance and cooperation" (PK 2015, 19; PK 2018, 22), of "voluntary subordination of personal interests to common goals" (PK 2015, 19), and a society without conflicts of interest among its constituent groups (elite and people) (PK 2017, 21). Kirill says:

> "Our ideal / ... / is a solidary society, a society of social symphony, where different strata and groups, different peoples and religious communities, different participants in political and economic processes are not competitors fighting each other, but co-workers.". (PK 2013, 17)

Additionally, when at some instances Kirill presents lists of "high spiritual ideals of historic Rus'" (PK 2014, 18) and basic or fundamental values of national identity (PK 2011, 15), his lists include 'solidarity', but not *sobornost'*. Despite changes in the use of terms, his core message is still about *sobornost'*. According to Kirill, there exists a historical, spiritual, religious, Orthodox 'cultural code', which cannot be abandoned or threatened by social reforms and political decisions:

> "[T]he thousand-year history of Russia has formed a powerful spiritual and cultural code of our people, which directs the way of life of an individual and the whole society. . . . This code cannot be destroyed, because it is a set of truths that were formed under the influence of religious tradition and have been tested by the experience of people's life. . . . reforms in our country should not encroach on the cultural code of Russia.". (MK 2007, 11)

The 'normative order' of Russian culture can be referred as a 'cultural code' or 'solidarity society', but it retains also the defining opposites of sobornost'. The antipodes of 'solidary society' are chaos and collapse (PK 2012, 16), individualism (PK 2013, 17), society of (permanent) conflict, struggle and competition (PK 2013, 17), of aggressive secularism, 'fierce competition', and 'natural selection' (PK 2018, 22).

2.4.3. Sobornost' at the Level of Empire

*Sobornost'* (of the Slavophiles) is focused on the level of the Russian village. However, the 'village level' *sobornost'* is virtually missing in this sample of speeches (this aspect will be assessed in detail in the Discussion section), while elements of the imperial *sobornost'* (or *sobornost'* at the level of the empire) were systematically present in Kirill's speeches, despite the fact that he himself refrained from an articulation of its imperial features with the term 'imperial'.

In the first council, Kirill denounced the internationalism of the Soviet Union, because it put the Russian people in a "downtrodden position", and into "humiliated situation", where the Russian people (народ) largely lost its "self-identification" (MK 1993, 1). The natural principle for Russia to follow is universalism, which puts "forward the Russian nation as the spiritual leader of the oekumene (ойкумена)", and Kirill admits that this strategy risks being perceived as a kind of imperial ambition for Russia conceived of

by peoples of the former USSR (MK 1993, 1). In this regard, it is important to note that Metropolitan Kirill prefigures the arguments of Russian Orthodox domination of the "Russian World" (*Russkii mir*). He worries about the nationalism primarily of the "small" and "medium-size peoples" "who make up Russia (former USSR)", as if these smaller nations can be taken for granted as also "Russian". Although he rejects the charge that Russia sees these smaller nations as "little brothers", that is, in fact, the way the Russian World concept functions.

Speeches of both Patriarchs contribute to a discourse about imperial identity. In the sessions of the WRPC, the concept of empire was rethought (MK 2007, 11). On several occasions Kirill responded to criticisms of Russia having imperial aspirations with messages about 'common values' that unite peoples of historic Russia (MK 2004, 8; PK 2014, 18). Essentially, he responded to criticisms of Russian imperial ambitions with appeals to the imperial characteristics of Russian civilization (e.g., common spiritual values, history, and fate). The Ukrainian events of 2014 (Maidan, annexation of Crimea, military conflict in Donbas) are seen as: "Such a tragic division, which occurs when people lose a common understanding of their history, leading to division and provoking civil conflict, we are witnessing today in Ukraine" (PK 2014, 18).

For Kirill, the political and cultural achievements of the historic Russian Empire present a model to follow (PK 2014, 18); the ideals of 'solidary society' are the social ideals of Russia due to the civilizational choice of Prince Vladimir, who, in turn, by his choice created the unique Russian civilization (PK 2015, 19). Russia should be a "single family of many nations" (PK 2009, 13). Russian civilization includes many religions, where Islam, Judaism, and Buddhism are integrated into the civilization by shared values of a 'solidary society' (PK 2015, 19).

Russian civilization includes Ukrainians and Belarusians (PK 2019, 23), who are heirs of historical Rus', and the ROC sustains the unity of these peoples:

> "I am convinced that the values of our civilization are preserved among all peoples—the heirs of historical Rus': Ukrainian, Belarusian, Moldovan and many others. The Russian Orthodox Church has made and continues to make its contribution to the preservation of the historically formed unity of these peoples, to their mutual assistance and cooperation. We have a common spiritual root, common ideals, common goals—I am deeply convinced, and a common system of fundamental values, and therefore, of course, a common future.". (PK 2011, 15)

2.4.4. Advent of 'Nationalizing Imperialism' in a Multi-Religious and Multinational Empire

The Russian empire is multi-religious and multinational (consisting of many peoples), but particularly since 2014, Kirill has emphasized the special role of Russians and Orthodoxy in the Russian civilization. A defining feature of 'nationalizing imperialism' is its aim to establish a domination of single ethnicity and language within the territory it controls. Therefore, the litmus test for Russia is, to what extent in the territories that Russian Federation owns (and claims ownership over) it aims toward the establishment of the domination of ethnic Russians (Kolstø 2019, p. 19). Kirill's speeches since 2014 present evidence about a change in this direction.

Kirill claims that Russian people (*русский народ*) are "the most important subject of national relations in Russia", "genuine Russian (*подлинное русское*) national identity . . . does not threaten the integrity of Russia and interethnic peace in it, but on the contrary acts as the main guarantor of the unity of the country and friendship between its peoples" (PK 2014, 18). Russian people (*русский народ*) are the core around which "the Russian nation (*российская нация*), the Russian civilizational community, is being formed" (PK, 2014, 18). Russia became a historically great empire thanks to "the sacrifices made by the Russian people (*русским народом*)" (MK 2005, 9), together with the claim (this contradiction was discussed in the Section 2.1.5) that " . . . xenophobia and nationalism are incompatible with the very nature of Russia" (MK 2005, 9).

Hence, the Russian empire is simultaneously Orthodox—which itself is multi-ethnic and multinational (MK 1995, 2)—and multi-religious, multinational, and Russian:

> "/ . . . / Russian civilization is not only Russian and not only Orthodox-Christian, despite the decisive contribution of the Russian people and the Russian Orthodox Church to its creation. It is a common home and common heritage for peoples of different religions, faiths and cultures.". (PK 2015, 19)

Orthodoxy is of special importance to everyone in Russia, including non-Orthodox believers and non-believers. Peoples of Russia who follow other religious traditions are "rooted in Russian culture" and "are aware of the special importance of Orthodoxy in the formation of the national identity and spiritual identity of Russia" (PK 2022, 24). They are "able to reveal their identity and peacefully agree on the rules of joint life within the framework of a common multinational Russian civilization" (PK, 2014, 18).

Therefore, the symphony in Kirill's message is not so much between church and state, but in the social and cultural sphere of inter-ethnic and inter-religious relations. "The symphony of ethnic groups" gives Russian "civilization a unique appearance" (PK 2013, 17). Historical Russia aimed at 'harmony' in "a great Russian multinational world" (MK 2008, 12).

> "The identity of our country lies in the fact that it unites many peoples, cultures and religions. Thanks to the wise and balanced policy of our ancestors, they managed to create a unique polyphonic space of trust and cooperation, known as the Russian World, in which different peoples, cultures and religions peacefully coexisted, and which, unlike Europe, has never known wars on religious grounds.". (MK 2008, 12)

Of course, Russia also has national minorities, but these minorities strengthen the country economically, politically, and culturally, for all "the peoples of Russia, their country is their home, there is no other way. Therefore, every nation should feel truly at home, safe, in a benevolent environment of a multinational society" (PK 2014, 18).

Similarly, the Orthodox tradition "is culture-forming for Russian civilization" (MK 2006, 10). The territory of historic Russia "largely coincides with the canonical territory" of the ROC (Kirill 2018, 22). Patriarchs do not claim the territory of Georgia or Armenia as canonical for the ROC, but when the Russian state began its territorial expansion (in 15th century), Kirill claims that it was the ROC that expressed and sustained the universal ideal and mentality. With this reasoning, Kirill indicates that Orthodoxy made possible any territorial conquest of the historic Russian empire, including Georgia and Armenia. As a result, "[c]onsonant with the universality of the Church, the idea of a universal state created a state (*державу*)" (MK 1993, 1).

Thus, while Russia is a home of many peoples and religions, all peoples and religions of Russia recognize the special status of Orthodoxy and Russians:

> "But for Russia, the Russian world is not an ethnic concept. The Russian world includes all peoples belonging to other religions, but sharing the same values of social life together with the Russian people (*с русским народом*). It is Russia, which recognizes itself as Orthodox, that is able to maintain various cultures in unity. Over the centuries, Russia has developed a mechanism for the coexistence of different cultures and religions that accept the same social values, but retain their religious identity.". (MK 2004, 8)

> "The identity (*самобытность*) of our country lies in the fact that it unites many peoples, cultures and religions.". (MK 2008, 12)

Patriarch Alexy is less engaged with the thematics but, at the core his message agrees with that of Kirill: "The experience of Russia, where people of different faiths, different nationalities and cultural traditions have been living for centuries, is unique" (PA 2006, 10).

2.4.5. The Secular Dimension of Orthodox Trans-Nationalism

Importantly, the values of Orthodoxy that operate in Russian (trans-national) political culture do not distinguish and exclude those members of society who have no religious identity, belief, and affiliation. According to Kirill, the Orthodox Church supports and nurtures values such as "a strong and independent state, a free and responsible individual, an effective and fair economy, protection of the weak, a strong family, respect for the authorities, honest work" (MK 2004, 8). The values listed can be followed and acknowledged by religious and non-religious persons.

Therefore, true Christian patriotism should "not contrast religious and secular culture" and "does not exclude those who belong to non-Christian religions or those who seek moral motivation in secular humanism" (MK 1995, 2). Of course, there are also 'secular(ist)' values to be condemned and excluded—e.g., consumerism (PK 2014, 18), materialism (PK 2010, 14), egoism, and the "right of the strong" (MK 2001, 6).

2.4.6. Negative Nationalism Inside Multinational and Nationalizing Imperialism

In Kirill's view, the collapse of the Soviet Union was accompanied by the rise of negative nationalisms. He does not use the term 'negative nationalism', but he and Patriarch Alexy selectively endorsed some nationalisms of Orthodox peoples (such as Serbs and Bulgarians) (PA 2004, 8), while the (political) nationalisms of ex-Soviet peoples (particularly of Ukrainians) is denounced. The following statement is regarding the latter:

> "To elevate nationalism to the highest principle means to perpetuate and legit-
> imize the struggle that is tearing humanity apart, and to destroy the brotherhood
> of the peoples who made up and make up Russia . . . Our neighbors in the CIS
> have clung to nationalism as a level of self-awareness and self-realization. And
> in the face of these local nationalisms, Russians found themselves defenseless.
> The principle that binds the nation together has been lost. "Russians" became an
> empty sound, denoting nothing by name.". (MK 1993, 1)

In the Russian empire "the nature of the individual is supranational" (MK 1993, 1), while the Russian language and Russian people are still relevant. When Kirill says that Russians are "the most divided" nation of the world (PK 2018, 22), the reference is to the ethnic and/or linguistic identity of peoples living outside of the Russian Federation, who "found themselves outside historical Russia" as a result of the Civil War and the collapse of the Soviet Union, but who "belong to the same nation, one tradition, to which Russia is returning today" (MK 2004, 8).

*2.5. Russian Civilization, Globalization, and Confrontation with the West*

2.5.1. State-civilization discourse from 2009

Since the start of Kirill's patriarchy (in 2009), his civilizational discourse is consolidated on the notion of Russia as a state-civilization (PK 2013, 17; PK 2018, 22; PK 2019, 23):

> "Yes, Russia is a country-civilization, with its own set of values, its own laws of
> social development, its own model of society and the state, its own system of
> historical and spiritual coordinates.". (PK 2013, 17)

The civilizational identity of Russia is envisioned as being unchanged throughout history:

> "At any time, despite all the reforms, revolutions, counter-revolutions, Russia
> retained its civilizational basis. The models of the state structure, the titles of the
> rulers, the habits of the ruling classes changed, but Russian society, the Russian
> people retained their national identity.". (PK 2014, 18)

In particular, in 2013 Kirill said that when he speaks of Russia, he does not speak about the Russian Federation, but of Russia as civilizational space (including Ukraine) (PK 2013, 17).

However, before 2009, the civilizational discourses of identity used by Alexy and Kirill were different. Russia was also considered to be: one of the centers of the Orthodox

civilization (PA 2001, 6; PA 2004, 8; PA 2006, 10); a "bridge" between Western and Eastern civilizations (MK 2001, 6); a space where western and eastern, northern and southern religions, cultures, and traditions communicated (PA 2001, 6).

### 2.5.2. Unique Russian Civilization in a Multipolar World Order

The status of Russia in a global world order is a theme introduced to these religious leaders' speeches only in the 21st century. The messages of Alexy and Kirill overlapped significantly: the world structure should be 'multipolar', and every nation should have the opportunity to retain its cultural, historical, religious, and 'primordial' (PA 2001, 6) traditions and values (MK 2001, 6; MK 2004, 8; MK 2005, 9; PA 2004, 8):

> "I believe that today Russia's mission in the XXI century is that our unique (*уникальный*) experience in building a single (*единого*) civilizational space on the basis of cultural and religious diversity contributes to the construction of a multi-structured (*многоукладного*) world in which various civilizational models would be included in harmonious and peaceful interaction.". (PA 2006, 10)

### 2.5.3. Globalization and Confrontation with the West

Alexy mentioned the uneven distribution of power and wealth in the global world as unjust (PA 2001, 6), but Kirill had already added a dimension of liberal vs traditional values by 2001:

> " . . . today it is the non-religious, de-ideologized liberal standard that is offered to the world community as a universal model of the arrangement of the life of the state and the individual. However, many peoples seek to defend the right to their own traditional way of life, considering it not the property of the past, but the basis for the future.". (MK 2001, 6)

However, it is important to recognize, that the 'problem of values' was also domestic up until the third presidential term of Vladimir Putin (started May 7, 2012). Kirill argued that in the realm of spiritual values of Russia, there still was a vacuum that needed to be (truly) filled (MK 2006, 10; PK 2011, 15).

It is also important to highlight that when religious and traditional values are used to unify and consolidate at the level of the (imperial) state, and at the level of the Russian state, Kirill affirmed that Orthodox cultural values do not alienate non-religious people, but at the global level, cultural unification is a problem, and there is no consensus to be built amongst religious and secular values. Kirill argues that (at global level) religious tradition is indispensable for morality and human rights:

> "Religious tradition thus contains a criterion for distinguishing between good and evil. From the point of view of this tradition, they cannot be recognized as the norm: mockery of the sacred, abortion, homosexuality, euthanasia and other types of behavior that are actively defended today from the standpoint of the concept of human rights. If a person does not see that he is committing a sin, then everything is allowed to him . . . the norms of religious tradition are more authoritative for the believer's consciousness than earthly laws." (MK 2006, 10)

Kirill had already denounced liberal individualism in 1993, but then he did not connect this theme to global geopolitics:

> "The end of the twentieth century reveals to us a frightening picture of / . . . / the overwhelming importance of fashion (not only in clothing, but also in lifestyle) challenges individuality; legalization and promotion of homosexuality, the practice of surgical sex reassignment, again fashion blurs gender identity; the dominance of a unified "technoculture", a uniform one, completely devoid of any national roots of "pop culture", template video products undermines cultural and ethnic identity." (MK 1993, 1)

In 2016, however, Kirill points the finger at the global promotion of liberal values toward the "godless (*обезбоженный*) and dehumanized civilization of the West." "[T]he

most acute conflict of our time is . . . the clash of a transnational, radical, secular globalist project with all traditional cultures and with all local civilizations" (PK 2016, 20). The creation of secular society and "expelling religion from public space" is "one of the most important basic principles of the new Western European and Western culture in general" (PK 2022, 24).

The theme of 'globalization' and confrontation with the West is articulated at length by Kirill alone. For him, globalization is also an ideology (PK 2018, 22), which in the form of 'globalism': justifies the unifying process of globalization, is "a non-religious doctrine" and promotes projects that "are directed against the institution of the family as a solid structure that preserves and transmits tradition" (PK 2022, 24). An alternative is "many globalizations" instead of the single Western model and forming a common moral consensus via a dialogue of cultural and religious traditions (PK 2018, 22). Kirill advocates for 'pragmatic globalization' (PK 2018, 22), where:

> "Each cultural and historical subject will be forced in his own tradition to look for the support necessary for development and movement forward, to look for his own model of modernization, the origins of his system of social institutions." (PK 2017, 21)

In Kirill's universe, equality is among (nation-)civilizations. Among the latter exists "the Divine world order", based on Christian principles, that excludes "any attempts at dictate and unilateral imposition of political norms and cultural standards" (PK 2016, 20).

## 3. Discussion

We discuss the findings from four perspectives: Kirill's contribution to Russian imperialism; Kirill's construction of *sobornost'* in comparison to Russian 19th century applications of the same concept; comparison of his culture of *sobornost'* with Roman Catholic synodality; and Kirill's entrepreneurial construction of (imperial) national identity from the perspective of glocalization.

**1. Kirill's contribution to Russian imperialism.** The identification with imperial Russia became prominent in Russia during the second period of Putin's presidency, when, together with an emphasis on Russia's uniqueness and of a dignified status vis-à-vis the West, imperial Russia was identified as a positive Historical Other (Hopf 2016, pp. 226–27). Thereafter, historical Rus' and Russian civilization (in singular) were often used as synonyms in attempts to construct unity with earlier periods of Russian history as well as to integrate peoples living in the space of the former Soviet Union (Kazharski 2020, pp. 24, 29–30).

Particularly since his entry into the office of Patriarch in 2009, Kirill has promoted "civilizational nationalism" (Verkhovskii and Pain 2015). Russia needed "the Russian religious root meanings of its imperial existence" on the basis of pro-Orthodox and pro-imperial values (Vertlieb and Faleris 2022, pp. 140, 153). Our findings indicate that Kirill's messages have contributed to the restoration of imperial self-identification in Russian culture. Based on the presumed (conciliar) intrinsic cultural unity of all peoples of historic Russia, Kirill has offered lists of values and norms that are culturally Orthodox, socially secular, and politically all-inclusive, that do not exclude any group, ethnicity, religion, or ideology from the harmonious imperial culture.

**2. Kirill's culture of *sobornost'* in comparison with the *narodnost'* and *obshchinnost'* versions of 19th century.**

*Narodnost'* (nationality) and *obshchinnost'* (communality) are concepts that are closely related to *sobornost'* (Akhiezer 1997, p. 11; Vujacic 1996, p. 149). For 19th century Slavophiles, *narodnost'* characterized the nation as a whole and the people as set apart from the government (Vujacic 1996, p. 118). In the official theory of nationality of the Russian empire, presented by the Minister of Public Education Sergei Uvarov's (1786–1855) famous triad 'Orthodoxy, Autocracy and Nationality' (*narodnost'*) the term described "the sum total of national heritage" (Whittaker 1978, p. 172).

In the sample of speeches used in this study, the term *narodnost'* occurs just once in a sentence of Nikolai Danilevsky (1822–1885) which was quoted by Patriarch Kirill in 2014. The term *obshchinnost'* is never used, although Patriarch Kirill occasionally uses the affiliated term *obshchnost'* (*общность*) when he says that the WRPC should transform people into "a single community" (*в единую общность*) (MK 1995, 3) and that true unity of people cannot be ensured by force alone, but is based on "the spiritual and moral community (*общность*) of people living in the country, the commonality (*общность*) of values" (PK 2014, 18).

The main reason for the terms *narodnost'* and *obshchinnost'* not being used could be because Kirill's *sobornost'* is not focused on the historic traditions of Russian village culture and that he refrains from being identified with the bottom-up understandings of *sobornost'* (characteristic of 19th century Slavophiles). Kirill's *sobornost'* does not focus on the local level at all.

**3. Kirill's culture of *sobornost'* and Roman Catholic synodality.**

It would not be proper to compare the ROC and the Roman Catholic Church in a simplistic way, because our study was focused on the speeches of religious leaders withib the institution of the WRPC, which is essentially public and not exclusively representative of the ROC. Nevertheless, there is some profit from putting these two major churches side by side in a limited way and focusing on the similar ideas of *sobornost'* and synodality. With this is mind, we first compare Roman Catholicism with Orthodox Churches and then with the messages of two Patriarchs of Moscow.

A basic difference is in the ecclesiology of the Roman Catholic Church, which exists as a single global entity with national bishops' councils and the like, as compared with the Eastern Orthodox churches. Orthodox churches are in communion with each other, but the global system of association is based on ethno-national units that have their own autonomy or independence. This system is crystalized in the notion that each church has a primate, but there is no universal primate as in Catholicism's papacy.

The place of honor among all the patriarchs is occupied by the Patriarch of Constantinople, the Ecumenical Patriarch, but the Ecumenical Patriarch cannot control the activities or direct the policies of other patriarchs or church primates outside of his own church. He is *primus inter pares* or first among equals. Hence, the engagement of the Russian Patriarch (or bishops under the Russian Patriarch's jurisdiction) with Russian culture or civilization is not under the control or direction of any other church leader, and he is free to engage in moral entrepreneurship (Stoeckl 2016) of the type underlying the WRPC.

Our study findings demonstrate that as far as the public role of Orthodox values are concerned, Orthodox Russia is simultaneously Orthodox and multi-religious, Russian and multi-ethnic. Note that, therefore, there is a lack of equivalency of scale in comparing a church which is simultaneously one of the ethnonational churches of the Eastern Orthodox communion, but also the unique church of a multinational empire, with the global Roman Catholic Church, that is to say, the Russian Orthodox Church with the Catholic Church.

Nevertheless, there is value in exploring this comparison as it illuminates the conceptual foundations of *sobornost'*. It is noteworthy that Pope Francis and the global leadership of the Catholic Church are currently involved in a three-year process of exploring the idea of "synodality", and this concept is close to the Russian notion of *sobornost'* (albeit less with the top-down versions of it). According to the International Theological Commission of the Catholic Church, the term synod is associated with "the Tradition of the Church" and synodality with togetherness ("walking forward together") and "gathering in assembly". Synodality ought to be expressed in the Church's ordinary ways of living and working (Holy See Press Office 2021).

However, Roman Catholic synodality is predominantly a bottom-up process as envisioned by Pope Francis (Allen 2021). The *sobornost'* of patriarch Kirill is a top-down process conditioned by unity that precedes any decision-making and tends to exclude and eliminate the conflict of views, identities, and interests. Thus, while commitment to 'collegiality' is shared, and neither Catholics nor the Orthodox reject all types of the culture

of religious nationalism (Hollenbach 2022), the Roman Catholic approach lacks imperial-like commitments, an ideal of being in all-encompassing symphony with the holistically understood political culture, not to mention the legitimation of the war of aggression (Kilp and Pankhurst 2022a, 2022b).

**4. Kirill's entrepreneurial construction of (trans-national) national identity from the perspective of glocalization.**

Last but not least, there were surely glocal processes also involved in the attention paid in Russia to Orthodoxy following the demise of the USSR, the time when the state/society/nation/people were faced with the loss of national identity and belonging. Kirill must be credited with a clear moral entrepreneurial spirit to see, at the time, the need to deal with the challenges of identity on multiple levels and to initiate a process that was attempting to address these challenges in the form of elaborating what Russian nation, culture, and civilization means.

Glocalization (Roudometof 2013, 2014) entails experiencing the feedback from globalization processes that stimulates both the tendency to look for global patterns and sameness within which your culture/nation fits, but then to sense the need to make clear your national or cultural distinctness in various forms. In some measure, glocal stimuli must have been involved in Kirill's initiating steps to form the WRPC and work to sustain it for the decades subsequent to its founding.

Note that Kirill did not seek to establish a national church for Russia. He did not motivate people and groups only because they have thought of themselves as Russian citizens or as natives of Russia. Kirill's WRPC speeches have argued for a transnational Russian Orthodoxy, and, using Roudometof's (2014) terms, the approach is both nationalizing and trans-nationalizing, as the Church seeks to identify its own base nation alongside its international components. Arguing that, for example, Ukraine is part of Russia in some religious sense, as are out-migrant Russians, extends the current logic of glocalization for Russian Orthodoxy.

In lieu of the disintegration of the USSR, the appropriate anchor upon which one would center one's new project might not seem to be the floundering, weakened state(s) that emerged from the collapsed Soviet Union, abruptly losing power and prestige. A stronger foundation would seem to be the history, tradition, and widely recognized (and loved) culture of Slavic Russia which had had the Church at its core for centuries. The global status of that culture is unquestioned.

Furthermore, the emerging state system in which the Church found itself primarily located—the Russian Federation—did not seem sympathetic to the fate of its own Russian church. It had quickly declared religious freedom, an open market for religions, religions other than Orthodoxy rapidly filled the opening left by the collapse of the Soviet system (Pankhurst 1998), and democratic patterns were adopted as important goals toward which it was working. As we see from hindsight, these aspirations did not get institutionalized quickly and broadly enough to take root, but in 1989 and forward, the situation for the Church must have seemed daunting. Society needed some alternative focus to organize the chaos and to assuage its angst.

The great asset of the emphasis on *sobornost'* was and is that it is not limited by political boundaries or by the dimensions of the state or any other politically defined entity. One could argue that the message of the Church, that is, the message of universal salvation, should not, in any event, be seen as anything other than applying to unbounded humanity. That is the globalized idea of the Church. But, as with other dimensions of glocalization, the local demands definition and differentiation.

Basing your mobilization of people on the idea of *sobornost'*, then, provides the broad but differentiated foundation for that mobilization. The group is identified with an invisible spiritual entity that nevertheless has some boundedness, that is, the civilization or culture of Russia. Given the spread of Russian speakers and people born and raised in Russia, identifying with Russian culture means you belong to an international entity, that is, a

globally acknowledged "great" culture linked to world literature, music, dance and other performing arts, painting and other visual arts, and all the plastic arts.

In light of these dynamics affecting Russia at the beginning of the 1990s, the opportunities for social mobilization around the idea of *sobornost'* were ripe for Russians. As articulated by Metropolitan Kirill, the idea also echoed the basic tenet of the organization of the Russian Orthodox Church as worked out under the conditions of the 1917 Revolution but were then repressed by the Soviets. Major decisions about the direction of the Church were to be worked out in representative people's councils held in parishes and dioceses that would be headed periodically by a national (*vsenarodnyy, zemskii* or, ultimately *pomestnyi*) *sobor* which was to be made up of clergy, monastics, hierarchs, and laity (Pospielovsky 1984). This democratic structure led Petro (1995) to see the seeds of post-Soviet democracy for all Russia in the historical ROC patterns. Sadly, Kirill's original nominal inventiveness has been transformed into the neo-imperial (Chapnin 2015) and trans-national structures that have come to dominate ROC governance more recently.

## 4. Conclusions

When Russian president Vladimir Putin accused the Western nations of "moving toward open Satanism" in his 30 September 2022 speech (Kremlin 2022) marking the illegal annexation of four Eastern regions of Ukraine, it was not a first time that this secular leader would articulate a religious message (Drost and de Graaf 2022). Our data on the words of the primary post-Soviet Russian Orthodox religious leaders show the build-up of the interlinking of religious motifs with the political goals and achievements of the unfolding authoritarian regime of the Russian Federation over thirty years.

The findings of this study demonstrate significant agreement in the messages of Metropolitan Kirill, Patriarch Kirill, and Patriarch Alexy in the following themes: their general understanding of the culture of *sobornost'* in the historic space of the Russian empire, the promotion of Christian patriotism, the emphasis on the status of Russia as a great power in the multipolar global landscape, and on the rights of civilizations to follow their cultural values.

The main characteristic that distinguished Kirill's messages from the speeches of Patriarch Alexy was his habit to go several steps beyond Alexy both in content and style—both considered it important that Russia should follow its historic path, but only Kirill kept interpreting contemporary events with references to historic battles of Kulikovo, Borodino, and the Time of Troubles of 1612. Both appealed to a multipolar world order, but only Kirill identified the (Western) ideology of globalism as its driving force. Both encouraged people to serve the Fatherland, but Kirill says explicitly that the defense of the Fatherland may be more important than life.

**Author Contributions:** Conceptualization, A.K. and J.G.P.; methodology, A.K. and J.G.P.; selection of source speeches, A.K.; coding of speeches, A.K.; writing–original draft preparation, A.K. and J.G.P.; writing–review and editing, A.K. and J.G.P. All authors have read and agreed to the published version of the manuscript.

**Funding:** This research received no external funding.

**Institutional Review Board Statement:** Not applicable.

**Data Availability Statement:** The data for this article were the speeches given at the meetings of the World Russian People's Council given by Russian Orthodox Patriarch Alexy (1993–2008), and Metropolitan (1993–2008), then Patriarch (2009–2022) Kirill. The dates and themes of the meetings are listed in Appendix A, which includes the url addresses where the speeches are recorded in the meeting records. These are publically accessible at the websites indicated.

**Acknowledgments:** We gratefully acknowledge the comments of the anonymous reviewers and editors which have helped us improve this article.

**Conflicts of Interest:** The authors declare no conflict of interest.

**Appendix A  List of Council Years, Numbers, Titles and Transcript Sources**

1993 (1), 26–28 May. "Russian Councilial thought"/"Российская соборная мысль." https://vrns.ru/documents/stenogramma-i-vsemirnogo-russkogo-narodnogo-sobora/ (accessed on 30 April 2021).

1995 (2), 1–3 February. "Through Spiritual Renewal to National Revival"/"Через духовное обновление к национальному возрождению." https://vrns.ru/documents/stenogramma-ii-vsemirnogo-russkogo-narodnogo-sobora/ (accessed on 30 Apri l 2021).

1995 (3), 4–6 December. "Russia and Russians on the threshold of the XXI Century"/"Россия и русские на пороге XXI века." https://vrns.ru/documents/stenogramma-iii-vrns/ (accessed on 30 April 2021).

1997 (4), 5–7 June. "Health of the Nation"/„Здоровье нации." https://vrns.ru/documents/stenogramma-pervogo-dnya-zasedaniya-iv-vrns-zdorove-natsii/ (accessed on 30 April 2021).

1999 (5), 6–7 December. "Russia on the eve of the 2000th anniversary of Christianity. Belief. People. Power"/„Россия накануне 2000-летия Христианства. Вера. Народ. Власть." https://vrns.ru/documents/stenogramma-pervogo-dnya-zasedaniya-v-vrns-rossiya-nakanune-2000-letiya-khristianstva-vera-narod-vla/ (accessed on 30 April 2021).

2001 (6), 13–14 December. "Russia: Faith and Civilization. Dialogue of Epochs"/„Россия: вера и цивилизация. Диалог эпох." https://vrns.ru/documents/vi-vrns-rossiya-vera-i-tsivilizatsiya-dialog-epokh-13-14-12-2001/ (accessed on 30 April 2021).

2002 (7), 16–17 December. "Faith and Work: Spiritual and Cultural Traditions and economic Future of Russia"/„Вера и труд: духовно-культурные традиции и экономическое будущее России." https://vrns.ru/documents/vii-vrns-vera-i-trud-dukhovno-kulturnye-traditsii-i-ekonomicheskoe-budushchee-rossii-16-17-12-2002/ (accessed on 30 April 2021).

2004 (8), 3–5 February. "Russia and the Orthodox World"/„Россия и православный мир." https://vrns.ru/documents/viii-vrns-rossiya-i-pravoslavnyy-mir-3-5-02-2004/ (accessed on 30 April 2021).

2005 (9), 9–10 March. "Unity of peoples, unity of people—the key to Victory over fascism and terrorism"/„Единство народов, сплоченность людей—залог Победы над фашизмом и терроризмом." https://vrns.ru/documents/ix-vrns-edinstvo-narodov-splochennost-lyudey-zalog-pobedy-nad-fashizmom-i-terrorizmom-9-10-03-2005/ (accessed on 30 April 2021).

2006 (10), 4–6 April. "Faith. Person. Earth. Russia's Mission in the XXI Century"/„Вера. Человек. Земля. Миссия России в XXI веке." https://vrns.ru/documents/x-vrns-vera-chelovek-zemlya-missiya-rossii-v-xxi-veke-4-6-04-2006/ (accessed on 30 April 2021).

2007 (11), 5–7 March. "Wealth and Poverty: Historical Challenges of Russia"/„Богатство и бедность: исторические вызовы России." https://vrns.ru/documents/xi-vrns-bogatstvo-i-bednost-istoricheskie-vyzovy-rossii-5-7-03-2007/ (accessed on 30 April 2021).

2008 (12), 20–22 February. "Future generations—the national heritage of Russia"/„Будущие поколения—национальное достояние России." https://vrns.ru/documents/xii-vrns-budushchie-pokoleniya-natsionalnoe-dostoyanie-rossii-20-22-02-2008/ (accessed on 30 April 2021).

2009 (13), 21–23 May. "Ecology of the soul and youth. Spiritual and moral causes of crises and ways to overcome them"/„Экология души и молодежь. Духовно-нравственные причины кризисов и пути их преодоления." https://vrns.ru/documents/xiii-vrns-ekologiya-dushi-i-molodezh-dukhovno-nravstvennye-prichiny-krizisov-i-puti-ikh-preodoleniya/ (accessed on 30 April 2021).

2010 (14), 25–26 May. "National Education: Formation of an Integral Personality and Responsible Society"/„Национальное образование: формирование целостной личности и ответственного общества." https://vrns.ru/documents/xiv-vrns-natsionalnoe-obrazovanie-formirovanie-tselostnoy-lichnosti-i-otvetstvennogo-obshchestva-25-/ (accessed on 30 April 2021).

2011 (15), 25–26 May. "Basic values—the basis of the unity of peoples"/„Базисные ценности—основа единства народов." https://vrns.ru/documents/xv-vrns-bazisnye-tsennosti-osnova-edinstva-narodov-25-26-05-2011/ (accessed on 30 April 2021).

2012 (16), 1–2 October. "Frontiers of History—Borders of Russia"/„Рубежи истории—рубежи России." https://vrns.ru/documents/xvi-vrns-rubezhi-istorii-rubezhi-rossii-1-2-10-2012/ (accessed on 30 April 2021).

2013 (17), 30–31 October. "Russia as a country-civilization. Solidarity Society and the Future of the Russian People"/„Россия как страна-цивилизация. Солидарное общество и будущее российского народа." https://vrns.ru/documents/xvii-vrns-rossiya-kak-strana-tsivilizatsiya-solidarnoe-obshchestvo-i-budushchee-rossiyskogo-naroda-3/ (accessed on 30 April 2021).

2014 (18), 11 November. "Unity of History, Unity of the People, Unity of Russia"/„Единство истории, единство народа, единство России." https://vrns.ru/documents/xviii-vrns-edinstvo-istorii-edinstvo-naroda-edinstvo-rossii-11-11-2014/ (accessed on 30 April 2021).

2015 (19), 9–10 November. "The Heritage of Prince Vladimir and the Fate of Historical Rus'"/„Наследие князя Владимира и судьбы исторической Руси." https://vrns.ru/documents/xix-vrns-nasledie-knyazya-vladimira-i-sudby-istoricheskoy-rusi-9-10-11-2015/ (accessed on 30 April 2021).

2016 (20), 1 November. "Russia and the West: Dialogue of Peoples in Search of Answers to Civilizational Challenges"/„Россия и Запад: диалог народов в поисках ответов на цивилизационные вызовы." https://vrns.ru/documents/xx-vrns-rossiya-i-zapad-dialog-narodov-v-poiskakh-otvetov-na-tsivilizatsionnye-vyzovy-1-11-2016/ (accessed on 30 April 2021).

2017 (21), 1 November. "Russia in the XXI century: historical experience and development prospects"/„Россия в XXI веке: исторический опыт и перспективы развития." https://vrns.ru/documents/xkhi-vrns-rossiya-v-xxi-veke-istoricheskiy-opyt-i-perspektivy-razvitiya-1-11-2017/ (accessed on 30 April 2021).

2018 (22), 1 November. "25 years along the path of public dialogue and civilizational development of Russia"/„25 лет по пути общественного диалога и цивилизационного развития России." https://vrns.ru/documents/xxii-vrns-25-let-po-puti-obshchestvennogo-dialoga-i-tsivilizatsionnogo-razvitiya-rossii/ (accessed on 30 April 2021).

2019 (23), 18 October. "People's Saving—Present and Future of Russia"/„Народосбережение—настоящее и будущее России." https://vrns.ru/documents/xxiii-vrns-narodosberezhenie-nastoyashchee-i-budushchee-rossii/ (accessed on 30 April 2021).

2022 (24), 24 October. "Orthodoxy and the World in the XXI Century"/„Православие и мир в XXI веке." https://www.vrns.ru/documents/doklad-svyateyshego-patriarkha-kirilla-na-plenarnom-zasedanii-xxiv-vsemirnogo-russkogo-narodnogo-sob/ (accessed on 12 December 2022).

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
