# Peer review of "The Role of Moscow Patriarchs in the Promotion of the Imperial Culture of Sobornost’: Thematic Analysis of Religious Leaders’ Speeches at the World Russian People’s Council 1993–2022"

_religions, doi:10.3390/rel14040436_

Round 1

Reviewer 1 Report

The article is interesting and undoubtedly worth publishing. Based on an analysis of the statements of Patriarch Alexy and Metropolitan/Patriarch Kirill, it shows how the Russian Orthodox Church (ROC) uses the idea of sobornost’ to promote Russian imperialism and nationalism. I have a very favourable opinion of the article, but at the same time, I would like to bring a few things to the attention of the authors:

1. In the introduction, the authors define sobornost’ (pp. 1–2). They refer both to the ideas of the Slavophiles and to the position of contemporary Orthodox theologians. However, I think this description lacks criticism. Sobornost' is a ‘fake’ idea, Khomyakov's utopian dream. In reality, it was never realised. Slavophiles generally had little influence on state-church relations in the Russian Empire in the 19th century. Contemporary sobornost’ is merely a facade behind which a rejection of Western individualism (including the idea of human rights, the right to personal freedom, and freedom of conscience) and increasingly collectivism is hidden.

2. The authors do not pay proper attention to the particular form of Russian imperialism, which is Russia’s takeover of the history of the Rus’. Russia does not have a thousand-year history. Nor was St. Vladimir in any way connected to Russia. Great care must be taken in the selection of terms to avoid supporting language that falsifies history.

3. I strongly recommend that the authors abandon the translation of «Rus’» (Русь) as Russia. ‘Patriarch Kirill and All Russia’ (vv. 20, 21) , ‘Holy Russia’ (v. 333), ‘in the heart of historical Russia – in Ukraine’ (vv. 615-616), ‘heirs of historical Russia” (v. 788) – these expressions are incorrect. We know, of course, that the national separateness and sovereignty of Ukraine and Belarus are contested in contemporary Russia, especially by Vladimir Putin. However, even in the language of the ROC, there is no mention of ‘Holy Russia’ or Kyiv as the heart of ‘historical Russia’. Such a translation, although sometimes found, is wrong and does not correspond to the language used by Patriarchs Alexei and Kirill or the ROC in general. The best translation should be «All Rus’» and «Holy Rus’», although the somewhat outdated forms «All Ruthenia» and «Holy Ruthenia» are also sometimes encountered.

4. In my opinion, the paragraph comparing sobornost’ with the Roman Catholic concept of synodality (v. 1033 ff.) does not substantively connect with the rest of the article. Apart from some wording, they are completely different concepts. Sobornost’, if we understand it realistically, is simply a mask for Russian anti-Westernism, anti-individualism, and collectivism. It has nothing to do with Catholic synodality.

5. I recommend that the authors be more critical of Kirill Gundyaev’s theses in the discussion and conclusions. It can be inferred from the article that Patriarch Kirill is in fact concerned with the idea of sobornost’. In fact, especially since 2014, his discourse has been closely dependent on the political situation in Russia. The political needs of the Russian power elite are the proper perspective for interpreting Kirill’s words.

6. Two minor remarks: v. 32: Birjyukov à Biryukov; v. 110: Kravchuk à Kravchuk.

Author Response

Response to Reviewer 1

Overarching our response we want to emphasize that our paper is a political science and sociological analysis, not a theological one.  We do not wish to wade into complicated theological arguments; instead, we take at first what the patriarchs/metropolitan say at face value and as they intend it according to our analysis.  The issue of sobornost’ is one of the most important and complex theological issues in Orthodoxy, and we focus on it as the vehicle for the patriarchs’ intent in the context of the 1990s to the present.  Patriarchs Alexy and Kirill use the concept as a symbol of interpretation, and we want to understand their meaning and interpretation as they sought to convey it. Ours is a narrative analysis seated in social science of contemporary affairs with respect to theology and history, but not providing definitive definitions of either except as they are used by our two primary subjects, Patriarch Alexy and Metropolitan/Patriarch Kirill.

We have added a new opening paragraph to clarify our work and address the comments of reviewers.  We also provide point-by-point responses below. Thank you very much for your comments.

  1. We think, with the caveat stated above, we are not able to go further in the history of the concept of sobornost’ or its theological status in general. Surely, our interpretation is critical of the use of the concept by the patriarchs and, in particular, Kirill’s manipulation of it for his own purposes is irregular from a simple Christian/religious point of view.
  2. We are very grateful for the caution about uses of Russia vs. Rus’ and their proper attributions in discussing Russian and Ukrainian history and culture. The separate paths of the two are becoming more and more clear to all scholars today, and we have tried to argue here about the misuse of the ideas of Russian centrality by the patriarchs in their hegemonic discourse.
  3. It is interesting that on the ROC website where there are English translations, there often is the use of “Russia” for “Rus’”, but the more exact translation is surely preferred. Correction of Russia into Rus’, where appropriate. Highly appreciated feedback from reviewer. The corrections have been made in significantly more instances, the text and all quotes have been rechecked.
  4. As a socio-ecclesiological concept shaping social processes at all levels of a church, the comparison with a similar concept in Catholicism, one of the original Patriarchates, seems to provide helpful clarification. We explain that it is different in several ways, but the bottom-up ideal is quite different from the top-down pattern of Kirill’s sobornost’.
  5. This reviewer reduces sobornost’ to a preferred meaning, but that meaning goes beyond what is in the narratives of the patriarchs even as it may have an analytical dimension that is consistent with this reviewer’s – a very goal and intention of our article. We have indicated how Kirill has used the concept in his own ways, and that points toward the function the reviewer points to.
  6. Birjyukov changed to Biryukov; Krawchuk is the author’s real name.

We are grateful for the comments of the reviewer.  Thank you for helping us improve the paper.

Reviewer 2 Report

I think this is a very important study, one that is especially needed in light of Russia's attack on Ukraine.  It is very illuminating and would help to explain the current stance of Kirill in relationship to the war.  I only have 3 substantive comments for consideration.  1) the author discusses 'sobornost' at the start in a way that makes it seem that there is an established definition.  Furthermore, the author almost completely ignores the theological roots, use, and discussion on 'sobornost,' especially in Bulgakov, Afanasiev, Schmemann, and Meyendorff, to name a few.  I think the author needs to frame the first part more as a discussion of a contested term.  The author also needs to present the theological meaning of the term, if anything, to show how Kirill's use can be theologically countered.  Second, I understand why the author structured this theme thematically, but I wonder if it would be better just to do it historically, to show Kirill's progression, and then to provide a thematic summary.  I found myself a bit lost sometimes when trying to track Alexei and Kirill chronologically.  I think there is a progression in the rhetoric and the article could be clearer on this progression.  3)  I'm not convinced by the author's division of secular and religous values.  I understand why the author is making this distinction--in order to highlight how much Kirill's rhetoric progressed in a way as to embrace goals that have nothing to do with the church.   But Kirill is constantly criticizing 'secularism' and in Kirill's eyes, he is promoting Christian values.  Also, there is so much literature in critical theory that argues that Western secular values are really Christian values (Asad).  So, this part needs further nuance. 

Author Response

Responses to Reviewer 2

Overarching our response we want to emphasize that our paper is a political science and sociological analysis, not a theological one.  We do not wish to wade into complicated theological arguments; instead, we take at first what the patriarchs/metropolitan say at face value and as they intend it according to our analysis.  The issue of sobornost’ is one of the most important and complex theological issues in Orthodoxy, and we focus on it as the vehicle for the patriarchs’ intent in the context of the 1990s to the present.  Patriarchs Alexy and Kirill use the concept as a symbol of interpretation, and we want to understand their meaning and interpretation as they sought to convey it. Ours is a narrative analysis seated in social science of contemporary affairs with respect to theology and history, but not providing definitive definitions of either except as they are used by our two primary subjects, Patriarch Alexy and Metropolitan/Patriarch Kirill.

We have added a new opening paragraph to clarify our work and address the comments of reviewers.  We also provide point-by-point responses below. Thank you very much for your comments.

  1. As indicated above, for us to go into the complex theological and historiographical discussions of sobornost’ in a full way would take us far afield from our narrative analysis, which is already quite long and complex. We provide several sources that provide more in-depth consideration of the term on its own ground.  Our new opening paragraph sets the stage more directly for the narrative analysis and why it is critical in its own socio-political context of the Ukraine war.
  2. We are satisfied with the conceptual organization of the argument as it emphasizes what for our purposes is essential, all of it is dated appropriately.
  3. Where the “further nuance” asked for in the 3rd point takes us for our argument is not clear. We are again focusing on the Patriarchs’ narrative and its functions in the war setting.  Kirill certainly separates secular values from religious ones as the reviewer notes. 

We are grateful for the comments of the reviewer.  Thank you for helping us improve the paper.